# Development of Enzyme-Mediated Duplex Exponential Amplification Assay for Detection and Identification of *Meloidogyne enterolobii* in Field

**DOI:** 10.3390/microorganisms13061353

**Published:** 2025-06-11

**Authors:** Bingxue Sun, Bo Gao, Rongyan Wang, Shulong Chen, Xiuhua Li, Yonghao Dong, Juan Ma

**Affiliations:** International Science and Technology Joint Research Center on IPM of Hebei Province, IPM Innovation Center of Hebei Province, Key Laboratory of Integrated Pest Management on Crops in Northern Region of North China, Ministry of Agriculture and Rural Affairs, P. R. China, Plant Protection Institute, Hebei Academy of Agriculture and Forestry Sciences, Baoding 071030, China; sunbingxuechina@163.com (B.S.); gaobo89@163.com (B.G.); rongyanw@163.com (R.W.); chenshulong65@163.com (S.C.); lixiuhua727@163.com (X.L.); dongyonghao1119@163.com (Y.D.)

**Keywords:** *Meloidogyne enterolobii*, detection, isothermal amplification

## Abstract

The root-knot nematode *Meloidogyne enterolobii* has emerged as a devastating pathogen in global agricultural systems. Its geographic distribution is progressively expanding from tropical to temperate zones, leading to difficulties in discerning the symptoms it causes from those of congeners such as *M. incognita*. Currently, some molecular diagnostic technologies (e.g., qPCR) have been established for detecting *M. enterolobii*, but these methods fail to meet field-based detection demands due to their reliance on laboratory-grade thermocyclers. We thus developed a method for detecting *M. enterolobii* based on enzyme-mediated duplex exponential amplification (EmDEA) technologies to address this issue. The EmDEA detection method demonstrated strict specificity for the target species, showing no amplification in 13 non-target nematodes or host tissue samples. Sensitivity analyses revealed detection limits of 3.6 × 10^−4^ ng/μL (purified DNA), 1/1000 of an individual nematode (single-organism detection), 8.97 nematodes/g sweet potato, and 4.08 nematodes/100 g soil, achieving equivalent performance to qPCR. Field validation confirmed successful on-site detection, with significantly higher nematode loads in root tissues (50.41–97.62 nematodes/g) than in rhizospheric soil (1.07–1.28 nematodes/g). The established detection method employs a 42 °C isothermal amplification technology paired with a palm-sized thermal module, enabling field-deployable detection. Its unique duplex exponential amplification mechanism achieves threshold determination 10 cycles (~10 min) faster than conventional qPCR. When integrated with rapid DNA extraction protocols, the entire workflow is completed within 40 min, improving detection efficiency. This study provides a molecular tool for the precise monitoring of *M. enterolobii*, offering critical support for formulating targeted control strategies.

## 1. Introduction

The root-knot nematode *Meloidogyne enterolobii*, a highly pathogenic species, has evolved into a critical biological threat to global agricultural production systems [1,2]. This plant-parasitic nematode exhibits exceptionally broad host adaptability, infecting over 100 economically significant crops, including solanaceae vegetables (tomato and pepper), Cucurbitaceae crops (watermelon and cucumber), leguminous plants (common bean and soybean), and root and tuber crops (potato and sweet potato), as well as affecting ornamental species and various resistant rootstock varieties [3,4,5]. In recent years, this nematode has caused substantial economic losses in countries such as the United States, Mexico, and China due to its strong pathogenicity [6,7,8]. Notably, its geographic distribution is progressively expanding from tropical to temperate climate zones [9], leading to the symptoms it causes being confused with those of congeners such as *M. incognita*, including root gall formation, root system deformation, plant stunting, and biomass reduction [9,10]. This morphological confusion complicates the field identification of pathogen species, resulting in the inappropriate selection of chemical controls, the failure of resistant cultivars, and heightened management challenges. Consequently, there is an urgent need to establish precise and efficient detection technologies to inform targeted control strategies.

Conventional diagnostics rely on perineal pattern morphology, which is focused on morphological and morphometric characteristics, such as perineal patterns, dorsal striae, lateral lines, dorsal pharyngeal gland orifice, and stylet length [10]. Nevertheless, the accuracy of morphological identification is limited by the operator’s technical expertise, as well as the challenges posed by overlapping morphological traits and the quality of samples, all of which increase the risk of inaccurate identification. In response to these limitations, molecular diagnostic techniques have been developed, including PCR, quantitative PCR (qPCR), and droplet digital PCR (ddPCR) [11,12]. These technologies have also been implemented for the detection of nematodes including the soybean cyst nematode (*Heterodera glycines*), the potato rot nematode (*Ditylenchus destructor*), the pinewood nematode (*Bursaphelenchus xylophilus*), and foliar nematodes (*Aphelenchoides* spp.) [13,14,15,16].

The molecular detection of *M. enterolobii* commenced in 2010 [17,18]. Firstly, a multiplex polymerase chain reaction (PCR) assay which was able to distinguish *M. enterolobii* from *M. incognita* and *M. javanica* within 6 h was established, but it lacked quantitative capability [19]. Subsequently, an LNA-based quantitative assay targeting the mitochondrial cytochrome c oxidase subunit 1 (*cox1*) gene enabling quantitative detection in complex DNA matrices was established [18]. Shortly after, an absolute quantitative droplet digital PCR (ddPCR) method was established, demonstrating superior sensitivity and accuracy compared with real-time PCR for soil nematode egg quantification [20]. Most recently, a triplex ddPCR protocol achieved 99.7% accuracy in field validations for the simultaneous detection of *M. enterolobii*, *M. incognita*, and *M. arenaria* in co-infection scenarios [21]. Despite these significant advancements in the current molecular detection of *M. enterolobii*, some persistent challenges remain: (1) field application limitations due to large thermocycler dependency; (2) suboptimal detection efficiency with 2–4 h of processing time; (3) an elevated technical proficiency threshold requiring dual expertise in nematode morphology and molecular biology. Therefore, there is a critical need to develop field-deployable and high-efficiency molecular diagnostic methods.

At present, some isothermal amplification technologies, such as loop-mediated isothermal amplification (LAMP), recombinase polymerase amplification (RPA), and enzyme-mediated duplex exponential amplification (EmDEA), have been developed and successfully applied to various species in the field, enabling rapid detection within 60 min [22,23]. The EmDEA process involves two sequential reactions [24]: (1) the isothermal amplification of target DNA using thermostable DNA polymerase followed by (2) transcription mediated by T7 RNA polymerase to generate specific RNA sequences. These RNA sequences hybridize with fluorescently labeled probes to produce detectable signals. The subsequent cleavage of the DNA probe by duplex-specific nuclease (DSN) releases the amplification products, enabling cyclic rehybridization with new probes for iterative signal amplification. Through repeated fluorescent excitation cycles driven by individual nucleic acid molecules, this cascade reaction achieves significant enhancement in detection sensitivity. This technology has been applied to multiple species, such as *Radopholus similis* [25], *Ciborinia camelliae* [26], *Amaranthus palmeri*, *Rhaponticum repens*, and *Euphrosyne xanthiifolia* [24], motivating its novel application in *M. enterolobii* diagnostics.

In this study, we developed a field-deployable EmDEA assay for the rapid detection and quantification of *M. enterolobii* which requires minimal technical expertise. The assay was validated with laboratory experiments, and field and greenhouse sampling. The established method is anticipated to play a pivotal role in the effective management and containment of *M. enterolobii* infestations.

## 2. Materials and Methods

### 2.1. Nematode Isolation and Acquisition

*M*. *enterolobii* were collected from infected sweet potato fields in Guangdong Province, China. Additionally, other nematode species, including *Ditylenchus destructor*, *Pratylenchus scribneri*, *P. neglectus*, *P. coffeae*, *Steinernema littorale*, *Heterorhabditis beicherriana*, *M*. *incognita*, *M. hapla*, and *Heterodera avenae*, were obtained from laboratory cultures or isolated from field samples by using the Baermann funnel technique [13]. Species identification was confirmed through morphological analysis and molecular characterization.

### 2.2. DNA Extraction

Three extraction protocols were employed:(1)Preparation of DNA crude extract: For single J2 nematode DNA extraction, 5 μL of lysis buffer (Suzhou Jingrui Biotechnology Co., Ltd., Suzhou, China) was added to a 0.2 mL PCR tube. A single nematode was picked, placed in the lysis buffer, and then cut into two sections by using a syringe needle [14]. An additional 5 μL of lysis solution was added. The mixture was incubated at 95 °C for 10 min, and after cooling, the DNA crude extract of a single nematode was obtained.

For plant tissue/soil DNA extraction, 0.1–0.5 g of plant tissue/soil was cut and ground for 3 min by using a grinding rod or grinder. Then, 100 μL of lysis solution (Suzhou Jingrui Biotechnology Co., Ltd., Suzhou, China) was added, and the mixture was incubated at 95 °C for 10 min. After cooling, the DNA crude extract of the plant tissue/soil was obtained.

(2)Soil DNA extraction: Total DNA was extracted from 0.5 g soil by using the FastDNA™ Soil Genomic Kit (MP Biomedicals, Santa Ana, CA, USA), following the manufacturer’s protocol.(3)Nematode DNA extraction: Bulk J2 nematode DNA was extracted by using the TaKaRa MiniBEST Universal Genomic DNA Extraction Kit (Takara Bio, Beijing, China).

### 2.3. Primer Design

Forward, reverse, and RNA probe primers were designed according to a previous study based on the IGS interspecies variability and sequence (HQ896359.1) [24]. The primer design adhered to the following rules: the primer length was between 26 and 30 nt; the 5’ end of the RNA probe primer was labeled with a FAM probe and 5’rApp, and the 3’ end was labeled with a BHQ1 probe; the product length was less than 500 bp, preferably 100–200 bp. Primers demonstrating the lowest threshold cycle values (Ct) and the highest endpoint fluorescence intensities were identified as the optimal combinations. Specific primer information is provided in Table 1.

### 2.4. qPCR Detection Reaction

qPCR amplification was performed by using qPCR SuperMix (AQ601; Beijing TianGen Biotech Co., Ltd., Beijing, China). The 20 μL reaction mixture comprised 10 μL of 2× premix (final 1× concentration), 0.4 μL each of forward and reverse primers, 1 μL of template DNA, and nuclease-free water to final volume. The reaction procedure included pre-denaturation at 94 °C for 30 s, followed by 40 cycles of denaturation at 94 °C for 5 s, annealing at 64 °C for 15 s, and extension at 72 °C for 20 s. Finally, a melting curve analysis was performed.

### 2.5. EmDEA Reaction

Fluorescent isothermal amplification detection kits and RNA primer probes were purchased from Suzhou Jingrui Biotechnology Co., Ltd. The EmDEA reaction system (20 μL) included 1 μL each of forward primer, reverse primer, and RNA probe; 7 μL of template; 10 μL of activation solution (reaction buffer, dNTPs, and magnesium acetate; Suzhou Jingrui Biotechnology Co., Ltd., Suzhou, China); and enzyme dry powder (endonuclease, dNTPs, and polymerase; Suzhou Jingrui Biotechnology Co., Ltd., Suzhou, China; for details, see [24,25]). After the system was configured, it was centrifuged at 12,000 rpm for 30 s and mixed by shaking. The reaction mixture was placed in a qPCR detector (Suzhou Jingrui Biotechnology) and incubated at 42 °C for 30 min, with signals collected every minute. Alternatively, the PCR tube was observed under ultraviolet light for the detection of fluorescence signals.

### 2.6. Specificity Detection

Genomic DNA was obtained with crude DNA extraction from *D*. *destructor*, *P*. *scribneri*, *P*. *neglectus*, *P*. *coffeae*, *S*. *littorale*, *H*. *beicherriana*, *M*. *incognita*, *M*. *hapla*, and *H*. *avenae* to serve as templates for specificity detection based on the EmDEA system. If non-target DNA shows no amplification or signal intensity is less than or equal to the negative control, specificity is confirmed; otherwise, it is absent. Three consistent experimental replicates confirmed result reliability.

### 2.7. Sensitivity Detection

DNA sensitivity: DNA was diluted in 10-fold gradients to different concentrations for EmDEA and qPCR detection. Result accuracy was confirmed only when three independent experimental replicates yielded consistent results. Sensitivity in soil: Different numbers of J2 nematodes were artificially added to approximately 0.5 g of soil, and DNA was extracted by using the FastDNA Soil Genomic Kit. The DNA was diluted to different concentrations for EmDEA and qPCR detection.

Single-nematode sensitivity: A single nematode was placed in lysis solution, cut into two sections, and DNA was extracted by using the crude extract method. The DNA was then diluted to different concentrations for EmDEA and qPCR sensitivity detection.

Sensitivity in plant tissue: Different numbers of J2 nematodes were mixed with approximately 0.03 g of sweet potato tissue, and DNA was extracted by using the crude extract method. The DNA was diluted to different concentrations for EmDEA and qPCR sensitivity detection.

### 2.8. Validation of EmDEA Assay in Naturally Infected Plants

To further validate the reliability of the EmDEA assay, twelve samples were collected from diseased plants. DNA was extracted by using the crude extraction method and the FastDNA Soil Genomic Kit. The extracted DNA was subjected to EmDEA and qPCR. Each sample was tested in triplicate.

## 3. Results

### 3.1. Specificity

EmDEA detection showed significant specificity when used for amplifying and detecting DNA from 14 common nematodes and sweet potato tissues. Only the DNA sample of *M. enterolobii* produced an amplification signal, while no amplification was observed for the other nematodes or plant tissues (Figure 1), confirming the high specificity of this essay.

### 3.2. Sensitivity

The sensitivity of the EmDEA detection system was evaluated by using nematode DNA, single nematodes, nematode-inoculated plant tissue, and nematode-inoculated soil. The results are as follows.

DNA sensitivity: The detection limit was 3.6 × 10^−4^ ng/μL in the qPCR detector (Figure 2A). Under ultraviolet light, the sensitivity increased by an order of magnitude to 3.6 × 10^−5^ ng/μL (Figure A1). This might be due to the difference in the intensity of the fluorescence signal detected by the instrument and that obtained under ultraviolet light. A similar phenomenon was observed in the single-nematode sensitivity analysis. Under UV-based detection conditions, the method exhibited significantly higher sensitivity than the qPCR detection system, exceeding it by approximately one order of magnitude (Figure A2).

Single-nematode sensitivity: The detection limit was 1/1000 of a nematode (Figure 2B and Figure A2).

Plant tissue sensitivity: The detection limit was 8.97 nematodes/g sweet potato tissue (Figure 2C).

Soil sensitivity: The detection limit was 4.08 nematodes/100 g soil (Figure 2D).

### 3.3. Standard Curve

Standard curves were constructed by using DNA at different concentrations, mixtures of sweet potato and nematodes, and soil with varying nematode concentrations to quantitatively detect the number of nematodes in plant tissue/soil and the concentration of DNA. The equations were as follows:

DNA: y = −4.1872x + 17.16 (R^2^ = 0.9875) (Figure 3A).

Sweet potato tissue: y = −11.179x + 228.18 (R^2^ = 0.9918) (Figure 3B).

Soil: y = −0.0953x + 1.5605 (R^2^ = 0.9661) (Figure 3C).

### 3.4. Detection of Meloidogyne Enterolobii in Field Soil and Tissues

Twelve samples were detected by using both EmDEA and qPCR for comparative assessment to validate the stability of the EmDEA detection method. The nematode concentrations were determined as 46.00–124.57 nematodes/g tissue in host roots and 1.07–1.28 nematodes/g soil (Table 2), confirming the stability and field applicability of the EmDEA method. A comparative analysis revealed high consistency between the two detection methods when using DNA templates extracted with the FastDNA™ Soil Genomic Kit. However, when crude DNA extracts were employed as templates, qPCR exhibited CT values fluctuating between 32 and 40, demonstrating questionable reliability (Figure 2D).

### 3.5. Comparison Between qPCR and EmDEA Detection Methods

Sensitivity: The sensitivity of qPCR and EmDEA remained consistent when amplifying pure DNA templates extracted from pure nematode and soil samples (DNA extracted with the FastDNA™ Soil Genomic Kit). However, when using crude DNA extracts (derived from nematode-infected sweet potato tissues) as templates, EmDEA showed higher sensitive than qPCR (Figure 2).

Inhibitory effects of crude DNA extracts in qPCR and EmDEA: The crude DNA extracts (from nematodes) from simple matrices (e.g., purified nematodes) exhibited comparable amplification efficiency in both the qPCR and EmDEA systems (Figure 2B). However, as sample complexity increased, significant inhibitory effects emerged, disproportionately impacting qPCR. When using the crude DNA extracts (from plant tissues) from nematode-infected sweet potato tissues (moderate complexity), qPCR displayed elevated CT values (32–36) (Figure 2), indicating partial inhibition. In contrast, the EmDEA essay showed stable amplification under these conditions, demonstrating higher tolerance to moderate-complexity matrices. The inhibitory effects intensified in highly complex samples. For the crude DNA extracts from soil, the qPCR and EmDEA methods exhibited no amplification in soil samples, suggesting that humic acids and ionic contaminants critically impair enzymatic activity.

Detection efficiency: The EmDEA system significantly enhanced detection efficiency through workflow optimization in both purified and crude DNA workflows. For purified DNA templates, EmDEA reduced amplification time by 10 min (~10 cycles, at the same fluorescence brightness; Figure 2A–D) compared with qPCR on account of its dual-enhanced exponential amplification technology while eliminating the 30 min melt curve analysis, achieving a total time reduction of 40 min. When analyzing crude DNA extracts (from plant and root tissues), the method demonstrated further efficiency gains: simplified DNA extraction saved 30 min, accelerated threshold determination, and reduced detection time by 10 min, and the omission of the melt curve analysis eliminated an additional 30 min, culminating in a total time reduction of 70 min (Figure 2, Table 2 and Table 3).

Field adaptability: EmDEA detection requires the use of a portable amplification device enabling on-site field detection (Figure A3) or can be integrated with a compact metal bath for temperature regulation and a blue-light gel imager for the amplification analysis of field samples. Compared with the large-scale laboratory equipment required for qPCR, this system demonstrates enhanced field applicability.

## 4. Discussion

The root-knot nematode *M. enterolobii*, as a highly pathogenic plant-parasitic species, poses severe challenges to global agriculture due to its ongoing expansion [2]. Conventional morphology-based diagnostic approaches often prevent non-specialists from reliably differentiating *Meloidogyne* such as *M. incognita*, consequently inducing misdirected pest management decisions and compromised chemical control efficacy [28]. Although molecular diagnostic techniques like qPCR and ddPCR enable species-specific identification [17,21], their dependence on sophisticated thermocycling equipment and trained personnel significantly limits field applications. Thus, we developed an EmDEA-based detection system that enables field-deployable diagnostics with low technical barriers and miniaturized equipment to overcome these limitations.

In target sequence selection, the ribosomal intergenic spacer (IGS) sequence demonstrated superior interspecies polymorphism and intraspecies conservation compared with chloroplast genomes [29], with its multi-copy characteristic ensuring detection reliability [30]. The EmDEA system constructed from this sequence exhibited strict specificity, generating amplification signals exclusively for *M. enterolobii* with no amplification observed from 10 non-target nematode species. Sensitivity analyses on root tissues and nematodes confirmed equivalent detection sensitivity to qPCR. Field validation further demonstrated applicability in agricultural settings, revealing a significantly higher nematode number from infected root tissues than from rhizospheric soil.

Molecular detection methods encompassing conventional PCR, quantitative real-time PCR (qPCR), digital droplet PCR (ddPCR), and multiplex ddPCR assays have been systematically established for *M. enterolobii* [17,18,19,20,21]. However, field-deployable detection methods capable of being directly implemented on site remain underdeveloped. In this study, for the first time, we developed a field-deployable EmDEA detection method. Compared with existing detection systems (qPCR/ddPCR) for *M. enterolobii* [17,18,19,20,21], the field adaptability of EmDEA is highlighted by two key advancements: First, device miniaturization. The 42 °C isothermal amplification enables the use of palm-sized equipment, with the entire system being portable in a handheld case (Figure A3). Under extreme conditions, the amplification module can utilize a metal bath thermostatic unit. Second, detection time reduction. The EmDEA mechanism allows each amplicon to trigger multiple fluorescent signals, reducing threshold determination time by 25% (~10 cycles).

However, limitations persist. First, sensitivity in ultra-trace detection remains inferior to that of ddPCR, and absolute quantification remains unachievable [31]. For instance, ddPCR achieves detection limits as low as 0.9–1.4 copies/reaction in soil DNA due to its partitioning mechanism that isolates single molecules for amplification [32,33]. In contrast, EmDEA relies on bulk signal averaging, with detection limits of 5–10 copies, and necessitates calibration curves for semi-quantitation, thereby introducing variability risks [34]. Second, spatial heterogeneity in soil/root nematode distribution introduces variability in small-sample analyses (<1g). Increasing sampling points and compositing multiple subsamples reduces variability, mitigating this issue. Nevertheless, the performance of such small-scale methods is significantly lower than that of the Baermann funnel technique (100 g soil enrichment) in accuracy [35]. Third, within EmDEA field detection, current crude DNA extraction protocols demonstrate primary applicability in plant tissue samples (e.g., root tissues infected with root-knot nematodes) while exhibiting significant limitations in soil sample processing. This constraint likely stems from soil matrix interference factors, such as elevated humic acid concentrations, metal ions, and complex microbial consortia, which collectively inhibit DNA polymerase activity and disrupt fluorescence signal acquisition, ultimately leading to amplification failure. Three synergistic strategies may be employed to address soil-derived DNA amplification challenges: physical purification (SPE columns), contaminant sequestration (BSA/betaine), and enzymatic engineering (tolerant polymerase mutants) [36,37,38].

The molecular diagnostic method established herein exhibits significant potential for adaptation to other agriculturally relevant pathogens. Its versatility stems from two key advantages: (1) Cross-species applicability: It has been validated across taxa including nematodes (*R. similis* and *M. enterolobii*), fungi (*Ciborinia camelliae*), and invasive weeds (*Euphrosyne xanthiifolia*) [24,25,26], demonstrating cross-kingdom compatibility. (2) Field-deployable platform: Isothermal amplification enables rapid field detection, supporting the shift toward on-site diagnostics. Furthermore, we can leverage this detection technology to establish a molecular diagnostic system for nematicide resistance based on the rapid diagnosis of target gene mutations and multiplex resistance marker analysis [39,40]. This system will facilitate the development of regional resistance profiles to guide precision nematicide application strategies.

In conclusion, the EmDEA detection system bridges the gap between laboratory precision and field applicability on account of its low technical expertise requirements and miniaturized equipment. This provides a practical solution for real-time nematode diagnostics and establishes a methodological foundation for formulating precision management strategies against *M. enterolobii*.

## 5. Conclusions

We developed an isothermal amplification EmDEA assay for the rapid field detection and quantification of *M. enterolobii* requiring minimal technical expertise.

## Figures and Tables

**Figure 1 microorganisms-13-01353-f001:**
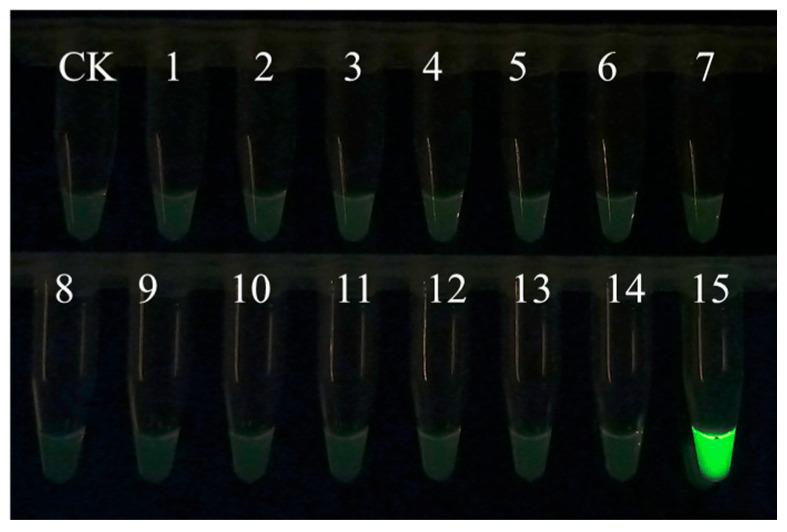
The specificity of the EmDEA assay. CK: negative control; 1: *Ditylenchus destructor* A population isolated from sweet potato in Hebei province, China; 2: *Ditylenchus destructor* B population isolated from angelica sinensis in Hebei province, China; 3: *Ditylenchus destructor* C population isolated from potato in Shanxi province, China; 4: *Ditylenchus destructor* from maize; 5: *Pratylenchus scribneri*; 6: *Pratylenchus neglectus*; 7: *Pratylenchus coffeae*; 8: *Steinernema littorale*; 9: *Heterorhabditis beicherriana*; 10: *Meloidogyne incognita*; 11: *Meloidogyne hapla* from pepper; 12: *Meloidogyne hapla* from sweet potato; 13: *Heterodera avenae*; 14: sweet potato; 15: *Meloidogyne enterolobii*.

**Figure 2 microorganisms-13-01353-f002:**
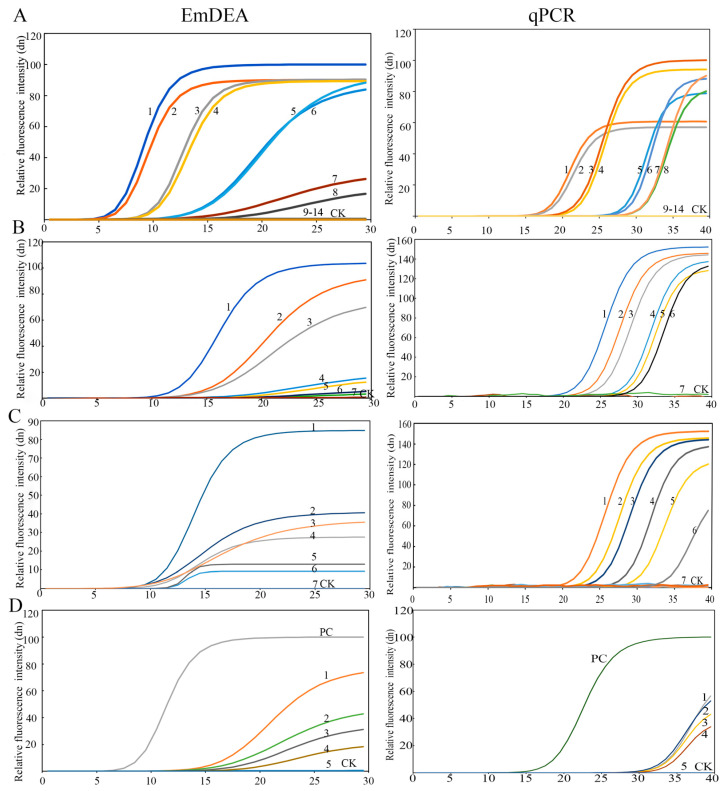
The sensitivity of the EmDEA assay and qPCR assay in different samples. (**A**), DNA sample: CK: negative control; 1 and 2: 3.6 × 10^−1^ ng/μL; 3 and 4: 3.6 × 10^−2^ ng/μL; 5 and 6: 3.6 × 10^−3^ ng/μL; 7 and 8: 3.6 × 10^−4^ ng/μL; 9 and 10: 3.6 × 10^−5^ ng/μL; 11 and 12: 3.6 × 10^−6^ ng/μL; 13 and 14: 3.6 × 10^−7^ ng/μL. (**B**), nematode sample: CK: negative control; 1: 1/10 of a nematode; 2: 1/50 of a nematode; 3: 1/100 of a nematode; 4: 1/500 of a nematode; 5: 1/1000 of a nematode; 6: 1/1500 of a nematode; 7: 1/2000 of a nematode. (**C**), soil sample: 1: 35.58 nematodes/100 g soil; 2: 20.85 nematodes/100 g soil; 3: 10.43 nematodes/100 g soil; 4: 9.98 nematodes/100 g soil; 5: 9.61 nematodes/100 g soil; 6: 4.08 nematodes/100 g soil; 7: 2.26 nematodes/100 g soil. (**D**), sweet potato tissue sample: PC: positive control; CK: negative control; 1: 67.59 nematodes/g sweet potato; 2: 45.32 nematodes/g sweet potato; 3: 13.27 nematodes/g sweet potato; 4: 8.97 nematodes/g sweet potato; 5: 4.23 nematodes/g sweet potato.

**Figure 3 microorganisms-13-01353-f003:**
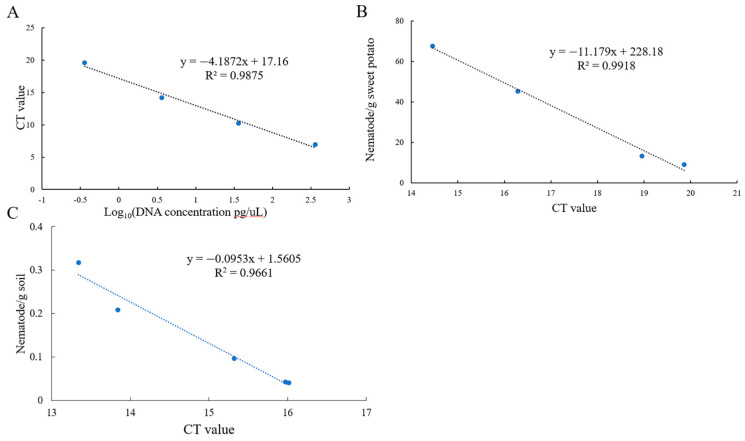
The standard curve of the EmDEA assay in different samples. (**A**): DNA sample; (**B**): soil sample; (**C**): sweet potato tissue sample.

**Table 1 microorganisms-13-01353-t001:** Primers in this study.

Primer	Sequence	Product	Origin
Me-F5	AAGCTAATACGACTCACTATAGGGTGTTGTT CGCTGTTCGCGGGAATGGTTT	241 bp	Designed in this study
Me-R1	AGTTTATTATTATAAGCTTTATTTTTTT		
Me-RNA1	UUCCUCCCACACAUUUUUAUAAUUGCCC		
Me-F	AACTTTTGTGAAAGTGCCGCTG	235 bp	[27]
Me-R	TCAGTTCAGGCAGGATCAACC		

**Table 2 microorganisms-13-01353-t002:** The results of the detection of *Meloidogyne enterolobii* obtained from the field using the EmDEA assay.

Sample ID	Host	Type	DNA Extraction Method	Origin	Concentration (Nematodes/g of Plant or Soil)	
					EmDEA	qPCR
FQ-1	Tomato	Root	Crude DNA extract	Baoding city, Hebei Province	53.74	/(32 < CT < 36)
FQ-2	Tomato	Root	Crude DNA extract	Baoding city, Hebei Province	46.00	/(32 < CT < 36)
YC-1	Tobacco	Root	Crude DNA extract	Baoding city, Hebei Province	108.32	/(32 < CT < 36)
YC-2	Tobacco	Root	Crude DNA extract	Baoding city, Hebei Province	69.29	/(32 < CT < 36)
YC-3	Tobacco	Root	Crude DNA extract	Baoding city, Hebei Province	124.03	/(32 < CT < 36)
YC-4	Tobacco	Root	Crude DNA extract	Baoding city, Hebei Province	123.57	/(32 < CT < 36)
YC-5	Tobacco	Root	Crude DNA extract	Baoding city, Hebei Province	121.49	/(32 < CT < 36)
GS-1	Sweet potato	Root	Crude DNA extract	Lufeng city, Guangdong Province	97.62	/(32 < CT < 36)
		Soil	Soil DNA extraction		1.28	1.35
		Soil	Crude DNA extract		No amplification	No amplification
GS-2	Sweet potato	Root	Crude DNA extract	Lufeng city, Guangdong Province	50.41	/(32 < CT < 36)
		Soil	Soil DNA extraction		1.07	1.21
		Soil	Crude DNA extract		No amplification	No amplification
GS-3	Sweet potato	Root	Crude DNA extract	Lufeng city, Guangdong Province	60.21	/(32 < CT < 37)
		Soil	Soil DNA extraction		1.27	1.52
		Soil	Crude DNA extract		No amplification	No amplification

**Table 3 microorganisms-13-01353-t003:** Comparison of qPCR and EmDEA detection methods.

Detection Method	Sensitivity				Detection Time ^5^	Field Operability	Professional Skills
	DNA ^1^	Individual Nematode ^2^	Sweet Potato ^3^	Soil ^4^			
qPCR	3.6 × 10^−4^ ng/μL	1/1500 of a nematode	8.97 nematodes/g sweet potato	4.08 nematodes/100 g soil	80~90 min	Not directly operable in the field	Strong
EmDEA	3.6 × 10^−4^ ng/μL	1/1000 of a nematode	8.97 nematodes/g sweet potato	4.08 nematodes/100 g soil	40 min	Directly operable in the field	Weak

^1^: DNA was extracted with the TaKaRa MiniBEST Universal Genomic DNA Extraction Kit. ^2^ and ^3^: Crude DNA was extracted with lysis buffer. ^4^: DNA was extracted with the FastDNA™ Soil Genomic Kit. ^5^: The detection time does not include the DNA extraction procedure.

## Data Availability

The original contributions presented in the study are included in the article, further inquiries can be directed to the corresponding author.

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
