# Peer review of "Development of Enzyme-Mediated Duplex Exponential Amplification Assay for Detection and Identification of Meloidogyne enterolobii in Field"

_microorganisms, 2025, doi:10.3390/microorganisms13061353_

Round 1
Reviewer 1 Report
Comments and Suggestions for Authors
The manuscript presents an innovative and relevant proposal by describing the development of an isothermal detection method (EmDEA) for Meloidogyne enterolobii, with significant potential for field application. The study is well conducted, with solid experimentation and validation in real samples, and contributes significantly to the advancement of rapid diagnostic tools in plant pathology.
However, some improvements are recommended to strengthen the clarity and applicability of the study:
Discussion of the limitations of the method in soil: The failure in detection with crude soil extracts should be discussed in greater depth. It is suggested to explore possible approaches to overcome inhibitory effects, such as the use of purification columns or contaminant sequestering agents.
Comparison with ddPCR: The manuscript could benefit from a more detailed discussion of the differences between EmDEA and ddPCR, even if based on the literature, especially regarding sensitivity and absolute quantification capacity.
Cost-effectiveness and scalability: Although the method is described as low-cost, it would be useful to include a comparative cost estimate per sample between EmDEA and qPCR, which would add practical value to the work.
Quality of the English language: A professional language review is recommended, as there are long sentences and grammatical structures that could be simplified to improve clarity.
Better integration of figures and appendices: Ensure that all referenced supplementary figures are properly explained in the main text and are easy to access and understand.
Future applications: Consider including a final paragraph highlighting the potential for adaptation of the method to other Meloidogyne species or pathogens, or its integration with nematicide resistance diagnostics. I congratulate the authors on their work and encourage publication after the suggested revisions.

The English used in the manuscript is generally understandable and adequately conveys the main ideas. However, there are sections with complex or unnatural grammatical constructions, as well as some punctuation and fluency errors, which may make it difficult for international readers to read. A language review by a native speaker or a professional scientific editing service is strongly recommended to improve the clarity, conciseness and consistency of the text.
Author Response
Comments 1:Discussion of the limitations of the method in soil: The failure in detection with crude soil extracts should be discussed in greater depth. It is suggested to explore possible approaches to overcome inhibitory effects, such as the use of purification columns or contaminant sequestering agents.
Response 1: Thank you for the suggestion. The discussion of limitations of method in soil was added in line 333-352. Modified content follows:
Third, within the EmDEA field detection, current crude DNA extraction protocols demon-strate primary applicability in plant tissue samples (e.g., root tissues infected with root-knot nematodes), while exhibiting significant limitations in soil sample processing. This constraint likely stems from soil matrix interference factors, such as elevated humic acid concentrations, metal ions, and complex microbial consortia, which collectively inhibit DNA polymerase activity and disrupt fluorescence signal acquisition, ultimately leading to amplification failure. Three synergistic strategies may be employed to address soil-derived DNA amplification challenges: physical purification (SPE columns), contaminant sequestration (BSA/betaine), and enzymatic engineering (tolerant polymerase mutants) [36-38].
Comments2: Comparison with ddPCR: The manuscript could benefit from a more detailed discussion of the differences between EmDEA and ddPCR, even if based on the literature, especially regarding sensitivity and absolute quantification capacity.
Response 2: Thank you for the suggestion. We have expanded the discussion in Lines 333-337 to explicitly compare EmDEA with ddPCR regarding sensitivity and quantification capabilities. Modified content follows:
However, limitations persist. First, sensitivity in ultra-trace detection remains inferior to that of ddPCR, and absolute quantification remains unachievable [31]. For instance, ddPCR achieves detection limits as low as 0.9–1.4 copies/reaction in soil DNA due to its parti-tioning mechanism that isolates single molecules for amplification [32,33]. In contrast, EmDEA relies on bulk signal averaging, with detection limits of 5–10 copies, and necessitates calibration curves for semi-quantitation, thereby introducing variability risks [34].
Comments 3: Cost-effectiveness and scalability: Although the method is described as low-cost, it would be useful to include a comparative cost estimate per sample between EmDEA and qPCR, which would add practical value to the work.
Response 3: Thank you for the suggestion. In our cost analysis, qPCR costs ¥20 per sample (¥13 for DNA extraction + ¥7 for amplification), while EmDEA costs ¥17.5 per sample (¥2.5 for DNA extraction + ¥15 for amplification). Given that the per-sample cost of EmDEA is only marginally lower than qPCR (¥2.5 reduction), we agree that describing EmDEA as "low-cost" relative to qPCR may overstate its advantage. Consequently, we have removed the term "low-cost" from the manuscript to avoid potential misinterpretation.
Comments 4: Quality of the English language: A professional language review is recommended, as there are long sentences and grammatical structures that could be simplified to improve clarity.
Response 4: Thank you for the suggestion. We have now engaged the professional language editing service recommended by MDPI to comprehensively refine the manuscript. Revisions are displayed with track changes.
Comments 5: Better integration of figures and appendices: Ensure that all referenced supplementary figures are properly explained in the main text and are easy to access and understand.
Response 5: Thank you for the suggestion. Referenced supplementary figures are properly explained in the main text in line 209, 216, 270 and 313. Modified content follows:
Line 209-216: DNA sensitivity: The detection limit was 3.6×10⁻⁴ ng/μL in the qPCR detector (Figure 2A). Under ultraviolet light, the sensitivity increased by an order of magnitude to 3.6× 10-5 ng/μL (Figure A1). This might be due to the difference in the intensity of the fluorescence signal detected by the instrument and that obtained under ultraviolet light. A similar phenomenon was observed in the single-nematode sensitivity analysis. Under UV-based detection conditions, the method exhibited significantly higher sensitivity than the qPCR detection system, exceeding it by approximately one order of magnitude (Figure A2).
Line 270: Field adaptability: EmDEA detection requires the use of a portable amplification device enabling on-site field detection (Figure A3), or can be integrated with a compact metal bath for temperature regulation and a blue-light gel imager for the amplification analysis of field samples.
Line 313: Compared with existing detection systems (qPCR/ddPCR) for M. enterolobii [17-21], the field adaptability of EmDEA is highlighted by two key advancements: First, device miniaturization. The 42°C isothermal amplification enables the use of palm-sized equipment, with the entire system being portable in a handheld case (Figure A3).
Comments 6: Future applications: Consider including a final paragraph highlighting the potential for adaptation of the method to other Meloidogyne species or pathogens, or its integration with nematicide resistance diagnostics.
Response 6: Thank you for this valuable suggestion. We have added a dedicated paragraph in the Future Applications section (Lines 353-362) to address the broader adaptability and diagnostic integration potential of our method.
The molecular diagnostic method established herein exhibits significant potential for ad-aptation to other agriculturally relevant pathogens. Its versatility stems from two key ad-vantages: (1) Cross-species applicability: It has been validated across taxa including nematodes (R. similis, and M. enterolobii), fungi (Ciborinia camelliae), and invasive weeds (Euphrosyne xanthiifolia), demonstrating cross-kingdom compatibility. (2) Field-deployable platform: Isothermal amplification enables rapid field detection, sup-porting the shift toward on-site diagnostics. Furthermore, we can leverage this detection technology to establish a molecular diagnostic system for nematicide resistance based on the rapid diagnosis of target gene mutations and multiplex resistance marker analysis [39,40]. This system will facilitate the development of regional resistance profiles to guide precision nematicide application strategies.
Comments 7: The absence of amplification with crude soil extracts is presented as a finding, but it would be interesting to know whether some type of internal control (e.g. spike with known DNA) was used to ensure that the failure was due to inhibition and not to the absence of nematodes or extraction failure.
Response 7: We sincerely appreciate this insightful suggestion. We will incorporate spiked internal controls in future experiments as recommended. In this study, although we did not use spiked known DNA as an internal control in the crude soil extracts—an oversight we acknowledge—we implemented the following validation steps to confirm that amplification failure was due to inhibition rather than nematode absence or extraction issues: (1) Identical DNA extraction methods were applied to plant tissues and soil samples. Plant tissue extracts consistently showed amplification curves, while soil extracts did not. (2) Experiments were repeated using artificially inoculated soil (with nematodes) and field-infected soil samples. No amplification was observed in either case. (3) We also asked the reagent manufacturer to test the same crude soil DNA extracts using our method. They got the same result—no amplification.
Comments 8: The variation in detected nematode numbers between root and soil samples is expected, but the manuscript should discuss the possible implications of spatial heterogeneity in nematode distribution and its impact on sampling protocols.
Response 8: We sincerely appreciate this insightful suggestion. As requested, we have added explicit discussion on spatial heterogeneity implications in the revised manuscript (Lines 339-342). The new text addresses two critical aspects:
Second, spatial heterogeneity in soil/root nematode distribution introduces variability in small-sample analyses (<1g). Increasing sampling points and compositing multiple subsamples reduces variability, mitigating this issue. Nevertheless, the performance of such small-scale methods is significantly lower than that of the Baermann funnel technique (100g soil enrichment) in accuracy [35]
Comments 9: The description of the reaction components could be more complete, especially regarding the function of the "enzyme dry powder". Would it be possible to include the enzyme composition or reference a published protocol?
Response 9: Thank you for this valuable suggestion. We have addressed this concern by adding the enzyme composition to the manuscript with corresponding supporting references (Lines 150-160). The revised text now reads:
Fluorescent isothermal amplification detection kits and RNA primer probes were purchased from Suzhou Jingrui Biotechnology Co., Ltd. The EmDEA reaction system (20 μL) included 1 μL each of forward primer, reverse primer, and RNA probe; 7 μL of template; 10 μL of activation solution (reaction buffer, dNTPs, and magnesium acetate; Su-zhou Jingrui Biotechnology Co., Ltd., Suzhou), and enzyme dry powder (endonuclease, dNTPs, and polymerase; Suzhou Jingrui Biotechnology Co., Ltd., Suzhou; for details, see [24,26]). After the system was configured, it was centrifuged at 12000rpm for 30 seconds, and mixed by shaking. The reaction mixture was placed in a qPCR detector (Suzhou Jingrui Biotechnology) and incubated at 42°C for 30 minutes, with signals collected every minute. Alternatively, the PCR tube was observed under ultraviolet light for the detection of fluorescence signals.
Comments 10: It is necessary to clarify whether the different “populations” of Ditylenchus destructor come from different regions, hosts, or simply represent biological repetitions.
Response 10: Thank you for this valuable suggestion. Different “populations” of Ditylenchus destructor come from different regions, hosts. The revised text now reads in line 194-200:
Figure 1. The specificity of the EmDEA assay. CK: negative control; 1: Ditylenchus destructor A population isolated from sweet potato in Hebei province, China; 2: Ditylenchus destructor B pop-ulation isolated from angelica sinensis in Hebei province, China; 3: Ditylenchus destructor C population isolated from potato in Shanxi province, China; 4: Ditylenchus destructor from maize.
Comments 11: The term “crude DNA extract” is widely used in the manuscript, but lacks precision. It is recommended to specify the type of matrix (soil, plant tissue or nematode) to which it refers in each instance, since the performance of the method varies according to this variable. E.g. lin238, line 246, line 254
Response 11: We appreciate this insightful observation regarding terminology precision. As suggested, we have systematically specified the biological matrix source for every occurrence of "crude DNA extract" throughout the manuscript. The modifications explicitly indicate whether extracts derive from soil, plant tissue, or nematodes at the following locations: Lines 260, 263, 264, 267, 272, and 281.
Comments 12:"To address these limitations, our study developed..." could be rephrased for style and conciseness. Minor typographical errors are also present (e.g., "ampli-fication" line breaks).
Response 12: Thank you for this valuable suggestion. We have addressed both points: (1) The suggested phrasing has been revised for improved style and conciseness in Line 307-309. The entire manuscript has undergone thorough language polishing. Proof of professional language editing is provided in the figure below.
Comments 13: achieving field-deployable diagnostics through significantly lowered technical barriers and miniaturized equipment." → rephrase to: "allowing field-deployable diagnostics with simplified operation and portable equipment."
Response 13: Thank you for this valuable suggestion. The suggested phrasing has been revised for improved style and conciseness in line 307-309. The entire manuscript has undergone thorough language polishing. Proof of professional language editing is provided in the figure below.

Reviewer 2 Report
Comments and Suggestions for Authors
The manuscript ""Development of Enzyme-mediated Duplex Exponential Amplification Assay for Detection and Identification of Meloidogyne enterolobii in Field" describes the development and validation of an enzyme-mediated duplex exponential amplification (EmDEA) assay for the rapid, field-deployable detection of Meloidogyne enterolobii. The work is timely and relevant, considering the increasing global spread and economic impact of this nematode species. The study addresses a significant limitation in nematode diagnostics—field applicability. The EmDEA approach is presented as a feasible, sensitive, and portable alternative to traditional lab-based molecular methods. The authors conducted comprehensive specificity and sensitivity analyses across multiple sample types (nematode DNA, plant tissue, and soil), validating the method through both lab and field trials. The side-by-side comparison with qPCR is especially valuable, highlighting EmDEA's advantages in terms of time efficiency, lower equipment requirements, and field adaptability. The manuscript is generally well-written and methodologically sound, with robust comparative analysis between EmDEA and qPCR.
Minor Comments:
While the authors acknowledge the poor performance of crude DNA extracts from soil in EmDEA, a more in-depth discussion or attempted optimisation (e.g., filtration or dilution protocols) would enhance the method's applicability.
The results are convincing, but the statistical methods used to analyze sensitivity, variability are not clearly stated. More extensive description of the statistical analysis would enhance the value of the publication.
Author Response
Comments 1: While the authors acknowledge the poor performance of crude DNA extracts from soil in EmDEA, a more in-depth discussion or attempted optimisation (e.g., filtration or dilution protocols) would enhance the method's applicability.
Response 1: Thank you for the suggestion. The discussion of limitations of method in soil was added in line 329-348. Modified content follows:
Third, within the EmDEA field detection, current crude DNA extraction protocols demon-strate primary applicability in plant tissue samples (e.g., root tissues infected with root-knot nematodes), while exhibiting significant limitations in soil sample processing. This constraint likely stems from soil matrix interference factors, such as elevated humic acid concentrations, metal ions, and complex microbial consortia, which collectively inhibit DNA polymerase activity and disrupt fluorescence signal acquisition, ultimately leading to amplification failure. Three synergistic strategies may be employed to address soil-derived DNA amplification challenges: physical purification (SPE columns), contaminant sequestration (BSA/betaine), and enzymatic engineering (tolerant polymerase mutants) [36-38].
Comments 2: The results are convincing, but the statistical methods used to analyze sensitivity, variability are not clearly stated. More extensive description of the statistical analysis would enhance the value of the publication.
Response 2: Thank you for your suggestion. Regarding specificity, sensitivity, and field detection tests, the entire study utilized three laboratory replicates per sample, and results were only accepted when all three replicates were consistent. As suggested, we have added an extensive description of the statistical analysis to the manuscript at the following locations: Line 167–169, Line 172–173,Line 189.
